# Benthic Diatom Communities in Korean Estuaries: Species Appearances in Relation to Environmental Variables

**DOI:** 10.3390/ijerph16152681

**Published:** 2019-07-26

**Authors:** Ha-Kyung Kim, In-Hwan Cho, Eun-A Hwang, Yong-Jae Kim, Baik-Ho Kim

**Affiliations:** 1Department of Environmental Science, Hanyang University, Seoul 04763, Korea; 2Department of Life Science, Daejin University, Gyeonggi 11159, Korea; 3Department of Life Science and Research Institute for Natural Sciences, Hanyang University, Seoul 04763, Korea

**Keywords:** CCA, cluster analysis, benthic diatom, random forest, estuary, diatom indices

## Abstract

In the Korean Peninsula’s southern estuaries, the distributive characteristics of epilithic diatoms and the important environmental factors predicting species occurrence were examined. The collection of diatoms and measurements of water quality and land-use were performed every May between 2009 and 2016, with no influence from the Asian monsoon and snow. Throughout the study, 564 diatoms were classified with first and second dominant species of *Nitzschia inconspicua* and *N. perminuta*. Based on diatom appearance and standing crops, the 512 sampling stations were divided into four groups by cluster analysis, and two regions, namely the West and East Sea. Geographically, G1, G2, G3, and G4 were located in the East Sea, Southeast Sea, West Sea, and Southwest Sea, respectively. Canonical correspondence analysis (CCA) results indicated that environmental factors, such as turbidity, electric conductivity (EC), and total phosphorus (TP), significantly influenced the distribution of epilithic diatoms. A random forest model showed that major environmental factors influencing the diatom species appearance included EC, salinity, turbidity, and total nitrogen. This study demonstrated that the spatial distribution of epilithic diatoms in the southern estuaries of the Korean Peninsula was determined by several factors, including a geographically higher tidal current-driven turbidity increase and higher industrial or anthropogenic nutrient-loading.

## 1. Introduction

An estuary is a transition zone where seawater and freshwater meet, and it is also a dynamic ecosystem with a diverse composition of living organisms due to significant physicochemical changes, such as water temperature, salinity, and nutrients [1,2,3]. Despite the fact that estuaries have various functions, including providing habitat, purifying water quality, and producing marine products, they are being destroyed by development projects concentrated in these regions [4]. As a result, nutrients, organic matters, and other pollutants are accumulating in coastal waters [5].

The Korean Peninsula is surrounded by seas on three sides, and due to an increase in coastline development relative to the narrow land area, approximately 460 estuaries have formed. Among these, only 235 estuaries are able to maintain estuarine circulation, whereas the estuarine circulation of the remaining estuaries is cut off by estuarine dams or sea dikes, which limit the formation of a brackish water zone that constitutes an estuarine ecosystem [6]. Moreover, development projects concentrated in estuarine regions destroy the various functions of estuarine wetlands, including providing habitat, purifying water quality, and producing marine products [4]. Meanwhile, estuaries in the Korean Peninsula exhibit different characteristics depending on their geographical location. In particular, estuaries located in the East Sea show the characteristics of high altitude, simple coastline, and clean sea area, while estuaries located in West and South Seas have characteristics of complex coastline, severe tidal range, and developed tidal flats. These estuarine watersheds consist mostly of forests (39.2%), farmland (28.8%), water areas (16.4%), and cities (5.8%) [4].

Generally, estuaries are characterized by the vigorous exchange or mixing of physicochemical factors on a regular basis due to freshwater and seawater [7,8,9]. Therefore, it is easy to assess the health of estuaries using epilithic biota, such as epilithic diatoms [10]. Epilithic diatoms, together with bacteria, act as primary bioelements involved in the immigration of aquatic substrates and formation of biofilm [11]. They are also important primary producers and a major food source for macro-invertebrates and fish in aquatic ecosystems [12,13,14]. Moreover, compared to other biota (benthic organisms and fish), epilithic diatoms are more responsive to changes in the physicochemical environment, such as luminous intensity, water temperature, salinity, and nutrients [15,16], and because of their low mobility, they are well known to be indicator organisms for identifying the cumulative effect of pollutants over a long period of time or predicting changes in high trophic level biota [17,18,19,20]. Based on these rationales, many countries have developed various indices using epilithic diatoms—Diatom Assemblage Index to organic water pollution (DAIpo) [21], Baltic Dry Index (BDI) [22], Toluene Diisocyanate Index (TDI) [23], and Indice de Polluo-Sensibilité Spécifique (IPS) [24]—and have used them to assess aquatic ecosystem health [25]. In Korea, aquatic ecosystem health assessments have been conducted every year since 2008, using epilithic diatoms for most of the rivers throughout the country [26]. However, there is very little information with respect to employing epilithic diatoms as indicator organisms for estuaries, as studies in Europe and Korea are very limited [27,28,29,30,31]. The majority of studies related to estuaries have concentrated on red-tide or eutrophication phenomena, which are caused by marine aquaculture or nutrients introduced into rivers, while other studies have focused mainly on ecological research of biological communities, such as phytoplankton, vegetation structure, fish, and benthic animals, including water quality [32,33,34,35].

Accordingly, the present study aimed: (1) to identify the physicochemical environment of estuaries and major environmental factors that impact the distribution and emergence of epilithic diatom communities; and (2) to compare the application of various existing freshwater epilithic diatom community indices used in Korea to assess estuary epilithic diatom community health.

## 2. Materials and Methods 

### 2.1. Study Area

The present study surveyed 512 estuarine zones in 317 rivers that are close to the ocean in Korea over a nine-year period (2009–2016). The selected sites are actually the total survey points designated by Ministry of Environment (MOE), Republic of Korea. Therefore, we selected all the river mouths (estuary) for the study. The surveys were conducted in May to avoid the influence of the heavy rainy season during summer. The 512 locations are not regularly and repeatedly sampled or surveyed, but only irregularly sampled in May during 2009–2016. In Korea, May has little or no impact of monsoon or rainfall against stream and estuary ecosystem. The survey sites were divided into three categories of eastern sea estuary (ESE; 135 sites), southern sea estuary (SSE; 230 sites), and western sea estuary (WSE; 147 sites) based on the nearby sea (Table 1 and Figure 1). The categorized estuaries were further divided into closed estuary (296 sites), where the estuarine circulation is cut off by estuarine dam or drainage gate, and open estuary (216 sites), where complete or partial estuarine circulation is taking place (Table 1). The sites were selected based on the 2008 Guide of MOE/National Institute of Environmental Research (NIER) [36], specifically as sites that form a boundary with the ocean and areas affected by the ocean in an upstream direction of the river.

### 2.2. Data Collection

#### 2.2.1. Water Quality

Water temperature, dissolved oxygen (DO), potential hydrogen (pH), electrical conductivity (EC), salinity, and turbidity of the survey sites were measured on-site using a multi water quality checker (Horiba U-50, HORIBA Ltd., Kyoto, Japan). The water samples needed for laboratory analysis were collected in a 2-L sterilized water collection bottle from each site and transported to the laboratory while being stored on ice. Biological oxygen demand (BOD) was calculated as the difference between the DO concentration measured on-site and the DO concentration of the water sample collected on-site in a 300 mL BOD bottle and subsequently incubated in an incubator for five days at 20 °C under dark conditions, in accordance with the Winkler azide method. Concentrations of dissolved inorganic matters were measured with the ascorbic acid method using a spectrophotometer (Optizen POP, Neogen Inc., Sejong, Korea), after the cadmium reduction method was used to determine total nitrogen (TN) and persulfate decomposition for total phosphorus (TP) [37]. For chlorophyll (Chl-a) concentration and ash-free dry mass (AFDM), three or more rocks, ≥10 cm in size with a flat surface, were collected from the survey sites. A soft brush was used to clean 25 cm^2^ of the upper section of the collected rocks, which were placed in plastic sample bottles using water from the site. The collected samples were kept under cold and dark conditions and transported to the laboratory for measurement by standard methods [37].

#### 2.2.2. Epilithic Diatom Community

The epilithic diatom samples that were scrubbed off with a soft brush from the upper part of the substrate collected from the survey sites were transported to the laboratory after fixing in Lugol’s solution, after which, they were washed using the permanganate method [38] and permanent samples were prepared using an encapsulant. Epilithic diatom samples were observed using an optical microscope (Nikon E600, Nikon, Tokyo, Japan) under 400× to 1000× magnification. The relative abundance of species present for the analysis of epilithic diatom communities was set to the number of diatom frustules being ≥500 under arbitrarily set microscopic field of view. Species were identified using the methods of Krammer and Lange–Bertalot [39,40] and classified according to the Simonsen’s classification system [41]. To determine the characteristics of epilithic diatom communities, the dominant species, dominance index [42], diversity index [43], richness index [44], and evenness index [45] of each survey site were derived.

### 2.3. Data Analysis

#### 2.3.1. Cluster Analysis

To characterize the distribution of epilithic diatom communities present in estuarine zones within Korea, a cluster analysis was performed based on the number individuals in epilithic diatom communities and number of species present. In the cluster analysis, species types were classified according to similarity between cluster composition using the Ward’s linkage method and Euclidean distance.

#### 2.3.2. Indicator Species Analysis

Indicator species analysis (ISA) was performed to determine the indicator species and indicator value (IndVal) of each group categorized by cluster analysis. The ISA is a non-hierarchical statistical analysis method that uses the relative abundance and frequency of each species at each survey site to calculate the IndVal, upon which, the indicator species is determined. In the IndVal method [46], IndVal appears within a range of 0–100 with higher values representing higher indicative power [47]. In this study, when the IndVal was ≥25, the species with IndVal that was five times higher than that of another group (good species) was selected as the indicator species for that group [48]. The Monte Carlo test was performed to determine the significance of the indicator species analysis.

#### 2.3.3. Random Forest

The random forest model was used to predict the presence of epilithic diatom species. The random forest model is a non-parametric method used to estimate and assess the relationship between latent predictor variables and response variables [49], using various combinations of environmental variables. The importance of environmental variables used in this model was determined using minimum description length (MDL), which is used for comparing the relative importance of environmental factors [50] with MDL values converted to a range between 0 and 100. To assess the predictive power of this model, the accuracy rate and area under curve (AUC) were derived. Accuracy was measured based on the dichotomous method of presence and non-presence, with a range of 0–1. The AUC could predict the reliability of the resulting values of the model and it typically has a range of 0.5–1, but values below this range may also appear.

### 2.4. Biological Integrity Assessment

Various epilithic diatom indices were derived and used for the biological integrity assessment of each survey site. The trophic diatom index (TDI) was used to assess the nutritional status of rivers by calculating the IndVal according to relative density, pollution sensitivity, and level of diatoms present in each site. This approach, together with diatom assemblage index of organic water pollution (DAIpo), has been utilized for a relatively long time [23]. As the nutrient concentrations increases, the value of this index increases, whereas when pollution levels increase, the value decreases. The present study used a modified version of TDI suited for Korea [51]. Meanwhile, DAIpo was used to calculate IndVal according to organic indicators (sensitive species and tolerant species) of species present in the survey sites [21], where an increase in level of pollution results in decrease in the value of this index. The *Achnanthes*/*Achnanthes* + *Navicula* (AAN) index is the proportion (%) of *Achnanthes* among total sum of *Achnanthes* (saproxenous species representing clear water areas) and *Navicula* (eutrophic species that prefer highly polluted water areas) [52]. An increase in the level of pollution results in a decrease in the value of this index. The motile diatoms (MD) index is a value that indicates the proportion (%) of motile diatoms in heavily polluted water areas among the total amount of diatoms present in each survey site [53]. An increase in the concentration of organic matters results in an increase in the value of this index. The classification of motile diatoms in this study followed the method by Hill et al. [54]. Lastly, the number of *Gomphonema* species (NGO) index is a value that indicates the percentage (%) of species that are member of the *Gomphonema* genus among all diatom species present in each survey site [52]. An increase in concentration of organic matters results in a decrease in the value of this index.

To compare the differences in community characteristics (total number of species and biomass), community indices, and environmental factors between the groups, Tukey’s post hoc test was performed for Analysis of Variance (ANOVA) nonparametric multiple comparisons. Moreover, Pearson’s correlation analysis was employed to analyze the relationships between indicator species and environmental factors for each group.

For cluster, indicator species, and Canonical Correspondence Analyses (CCA), the PC-Ord program (ver. 4.25. MjM Software, Gleneden Beach, OR, USA) was used. The random forest model was run by the CORElearn package in R statistic program (http://cran.r-project.org). ANOVA was performed using SPSS software (ver. 21. IBM, New York, NY, USA).

For epilithic diatom community data, species that appeared in less than 5% of all survey sites (25 sites) were identified as rare taxa and excluded from the statistical analysis. Moreover, to reduce variations among individuals, data were converted to the natural log (ln) and a value of 1 was added to each variable to prevent the ln value from becoming 0.

## 3. Results

### 3.1. Diatom Distribution and Community Characteristics

A total of 566 taxa of epilithic diatoms were present in 512 survey sites. The dominant species was *Nitzschia inconspicua* (13.9%), while the subdominant species was *Nitzschia perminuta* (7.4%). The total number of epilithic diatom species present was highest in the southern sea with 457 species, followed in order by the eastern sea (354 species) and the western sea (346 species). *Nitzschia inconspicua* was the dominant species in all of the sea areas, with the highest percentage of 18.9% found in the southern sea area (Table 2).

In the cluster analysis, using the current level of presence (cell density) of a total of 139 species after excluding epilithic diatoms that were present in <5% (<25 sites) of all survey sites, the species were grouped into four groups at a 25% level: Group 1 (G1: 91 sites), Group 2 (G2: 234 sites), Group 3 (G3: 89 sites), and Group 4 (G4: 98 sites) (Figure 2). G1, comprising mostly of sites located in eastern sea region, and G2, comprising sites in the southeastern coast of the Korean Peninsula, showed relatively high percentages of open estuaries with 96% and 71%, respectively. On the other hand, G3 and G4, mostly encompassing sites located in the western sea region, displayed high percentages of closed estuaries with 61% and 93%, respectively.

With respect to the dominant epilithic diatom species in each group, *Achnanthes minutissima* (8.5%) and *A. alteragracillima* (8.4%) were the dominant species in G1, but they showed very low percentages in other groups. On the other hand, *Nitzschia inconspicua*, which was the dominant species for all survey sites, displayed high relative frequencies in G2 (7.9%), G3 (19.4%), and G4 (11.5%), appearing as the dominant or subdominant species (Figure 3).

With respect to biological indices in each group, G3 had the highest number of species present, while G1 and G2 exhibited higher community indices and dominance indices than G3 and G4. Moreover, G3 and G4 had significantly higher diversity indices, G3 had the highest richness index, and G4 had the highest evenness index (*p* < 0.01; Figure 4).

In the indicator species analysis of each group targeting 139 taxa of epilithic diatoms with frequency ≥ 5% in all survey sites, there were 16 taxa with IndVal ≥ 25%, which were considered good indicators with values more than five times higher than other groups: G1 (5 species), G3 (8 species), and G4 (3 species). Meanwhile, no indicator species appeared in G2. The species with the highest IndVal (46%) was *Stephanodiscus invisitatus*, which was the indicator species of G3, but all other indicator species presented IndVal ≤ 40% (Table 3).

### 3.2. Physiochemical Water Quality

In the comparison of the physicochemical environment between the four groups (communities) categorized by distribution of epilithic diatoms using ANOVA, the factors that demonstrated significant differences were as follows (Figure 5): water temperature was lowest (18.5 °C) in G1 with highest percentage of ESE; pH (8.0) was highest in G3; and salinity (5.7 ppt) and EC (9396.8 μS/cm) were highest in G2 with highest percentage of ESE, SSE, and open estuaries (*p* < 0.01). Moreover, turbidity, TN, and TP were significantly higher in G3 and G4 than G1 and G2, while DO was highest in G1 and BOD was highest in G4 (*p* < 0.01). Consequently, the physicochemical environment of estuaries in the Korean Peninsula were classified similarly to the distribution characteristics of epilithic diatoms, except for water temperature, salinity, and EC that are affected by geographical influence. Furthermore, the results showed that water quality in the East-Southeastern Sea was better than that of the West-Southwestern Sea.

### 3.3. Relationship between Diatom Distribution and the Environment

In the correlation analysis between the indicator species and environmental factors of each group, most group indicator species displayed high correlations with turbidity, regardless of the groups, negative correlations with the indicator species of G1, and positive correlations with the indicator species of G3 and G4 (Table 4). In G1, *Cymbella silesiaca*, the species with highest IndVal in this group, showed a negative correlation with water temperature, pH, salinity, EC, turbidity, BOD, TN, TP, and AFDM, and a positive correlation with DO. In G3, *Stephanodiscus invisitatus*, the species with highest IndVal in this group, exhibited a positive correlation with pH, turbidity, TN, TP, and AFDM. In G4, the indicator species *Bacillaria paradoxa* correlated positively with turbidity, BOD, and AFDM. These results demonstrate contradicting correlations between the indicator species of G1 and the indicator species of G3 and G4.

The environmental factors that impact the presence of species in epilithic diatom communities were assessed using a random forest model. The results varied depending on the species, with accuracy ranging from 0.82 to 0.98 and the AUC displaying a range of 0.94 to 1.00 (Table 5). Among 139 taxa, the species that exhibited the highest predictive power was *Navicula atomus* var. *permitis* (accuracy: 0.98, AUC: 1.00), whereas the species with the lowest accuracy were *Gomphonema lagenula*, *Navicula gregaria*, and *Nitzschia dissipata*, and the species that displayed the lowest AUC value was *Achnanthes convergens*. To assess the contribution of environmental factors that impact the presence of epilithic diatoms, a sensitivity analysis was performed using MDL of random forest (RF). The most important factors that impacted the presence of epilithic diatoms were found to be EC (41 species, 29.5%) and salinity (36 species, 25.9%). These factors explained the presence of 55% of the species in epilithic diatom communities in estuaries. The results also showed that turbidity (13 species, 9.4%) and TN (13 species, 9.4%) were relatively important for the appearance of species in epilithic diatom communities (Table 4). Other important factors included EC for *Cymbella silesiaca* (indicator species with the highest IndVal in G1), pH for *Stephanodiscus invisitatus* (indicator species in G3), and BOD for *Bacillaria paradoxa* (indicator species in G4) (Table 4). Meanwhile, the main factor that determined the presence of epilithic diatoms in each group was water temperature for G1, salinity for G2 and G4, and TP for G3 (Figure 6).

### 3.4. Biological Integrity and Water Quality Assessment

When comparing the four groups by ANOVA using epilithic diatom indices for the assessment of biological integrity of estuaries in the Korean Peninsula, all items demonstrated significant differences (Figure 7). TDI, DAIpo, AAN, and NGO, which are items that decrease in value when the level of pollution increases, tended to be highest in G1 and lowest in G3. MD, which increases in value when the level of pollution increases, was lowest in G1 and highest in G3. Thus, the biological integrity was assessed to be high in G1, which exhibited low nutrient levels similar to water quality characteristics, while the integrity in G3 was determined to be low, with a relatively high level of nutrients.

## 4. Discussion

The present study analyzed the relationships between environmental factors and the distribution of diatom communities in estuaries in the Korean Peninsula. During the study period, 566 taxa of epilithic diatoms were identified in 512 sites. These results determined a higher number of taxa than the 327 taxa found in 161 sites between 2012 and 2014 in a previous study [31], which appears to be the result of the present study having more survey sites than previous studies. With respect to dominant species, the *Nitzschia* genus appeared in over 38% of estuaries in the Korean Peninsula. Moreover, the dominant species in the group that was directly affected by the ocean in the Ebro estuary of Spain included *Nitzschia frustulum* and *Nitzschia inconspicua* [55]. Meanwhile, the major epilithic diatoms present in estuaries in the Korean Peninsula (*Nitzschia*, *Navicula*, *Achnanthes*, and *Fragilaria* genera) have also been widely recorded in estuaries throughout the world, including Hungary, Sweden [56], the United States [57], Argentina, Uruguay [58], and the United Kingdom [59].

Based on the similarities of epilithic diatom communities in estuaries in the Korean Peninsula, a cluster analysis was performed to categorize four groups. G1 comprised the eastern sea region that included mostly the eastern area of Han River and parts of Nakdong River. G2 spread widely across the eastern and southern sea regions that included the eastern part of Han River, Nakdong River, and Seomjin River. According to Rho and Lee [4], land cover types in estuaries located in the eastern regions of Han River and Nakdong River showed a high percentage for forest (60%) and a low percentage for farmland (≤20%).

G1, located in eastern sea region, had low nutrient concentrations and significantly low levels of salinity and turbidity. This is consistent with previous studies where forest land use was reported to have a negative correlation with nutrient levels [60,61]. Furthermore, because there is a continuation of the ridgeline with rapidly descending altitude that extends from the Tae Baek Mountains to the East Sea, small rivers flowing into the East Sea have steep downward slopes and short extended waterways [62]. Tidal variations in this coast are minimal, and, thus, the tidal effects are usually very weak [63]. Accordingly, estuaries located on the eastern coast display lower salinity than those located on the western or southern coasts.

G3, located in the western sea region, included mostly the eastern section of Han River and parts of Geum River. G4, located in the western and southern sea regions, comprised mostly of the Geum and Yeongsan Rivers. The Geum River and Yeongsan River regions have proportions of farmland of ≥ 35%, which are much higher than the average of 28.8% for Korea. This is because of an increase in farming area from large-scale land reclamation by drainage and land clearing projects concentrated in these regions [4].

G3 and G4, located in western and southwestern estuaries, demonstrated significantly high levels of nutrients and turbidity. A previous study reported that a high percentage of farmland is highly correlated with total suspended solids (TSS) [64,65]. Moreover, when farmland is the type of land use, strong positive correlations with nutrients such as TN and TP are found [60], and as turbidity increases, DO concentration is known to decrease [66,67]. Furthermore, salinity and EC appeared at significantly high levels in G2, which appears to be the result of seawater flowing deeply into the estuaries due to the high percentage of open estuaries in the southern sea region.

*Achnanthes alteragracillima*, the subdominant species in G1, was almost not present in other groups, and this species is a saproxenous species that is present in relatively clean water. *Nitzschia inconspicua*, the dominant species in G2, G3, and G4, also appeared as the dominant species in 30 estuaries in 2012 [30].

*Cymbella silesiaca*, *Fragilaria rumpens* var. *fragilarioides*, and *Reimeria sinuata*, the indicator species of G1, correlated positively with DO and negatively with salinity, EC, turbidity, BOD, TN, and TP (Table 4). These species are typical saproxenous species [68], which are known to grow in oligotrophic and mesotrophic waters [69]. *Stephanodiscus invisitatus*, *Cyclotella atomus*, *Stephanodiscus hantzschii*, *Navicula veneta*, and *Navicula accomoda*, the indicator species in G3, displayed positive correlations with salinity, turbidity, TN, and TP and a negative correlation with DO (Table 4). These species grow mostly by floating in freshwater with high EC or are present in brackish water zones in rivers. They are known to be tolerant to organic pollutants [70,71]. The indicator species in G4, *Bacillaria paradoxa*, *Navicula capitate*, and *Nitzschia calida*, showed positive correlations with turbidity and BOD (Table 4). These species grow in brackish water zones or eutrophic waters with high EC, and they are known to have broad range of tolerance to pollutants.

Environmental factors that influence species presence or emergence in epilithic diatom communities were predicted using a random forest model (Table 5). The results showed that the most important factors were EC (41 species, 29.5%) and salinity (36 species, 25.9%). These factors explained the existence of 55% of the species in epilithic diatom communities in estuaries. Turbidity (13 species, 9.4%) and TN (13 species, 9.4%) also appeared as being relatively important for species presence in epilithic diatom communities. EC is known to be an important factor for determining the composition of epilithic diatom community [72,73].

In a preliminary study on diatom distribution associated with salinity, the salinity gradient was mostly the result of changes in the concentration of a single salt, sodium chloride (NaCl) [74,75,76]. As a result, it was difficult to differentiate the effect of a specific ion and the overall effect of osmotic pressure. Experimental results showed that medium osmotic pressure was an important factor in limiting the growth of freshwater diatoms [77] and affected nutrient intake [78]. Therefore, total ion strength and EC can explain the significant changes between diatom communities [72]. Amino acid, ammonium, and nitrate, which are a types of nitrogen compounds, are nutrients preferred by marine benthic diatoms [79]; however, because the present study only measured TN, it is necessary to conduct further experiments to measure a greater variety of nitrogen compounds. Therefore, if more detailed water quality (various nitrogen compounds and ions) measurements were conducted in the future, then more definitive evidence for distribution of epilithic diatoms might be available.

As a result of biological integrity assessment using epilithic diatom indices, the TDI could be divided into grades from A for very good to E for very poor, in accordance with the grading system given in the Biomonitoring Survey and Assessment Manual [51]. The TDI grade for estuaries was C (average) in G1, which had the highest IndVal of 42, while all other groups had a grade of D (poor). Compared to river water quality standards in Korea, G1 was assessed as having very good DO, somewhat good TP, and average BOD, with a TDI grade slightly lower than the water quality grade. Moreover, DAIpo, AAN, and NGO also showed low values below 50, relative to 100. This is because most of the epilithic diatom indices used in the present study were developed for freshwater systems, and thus, species that are ecologically important in estuaries were not included in the indices [80]. Moreover, some species that are included in the biological indices may not respond the same to environmental situations in estuaries and freshwater. For example, *N. frustulum*, which appeared as a major species with a high percentage of 6.7%, is very abundant in freshwater due to the high level of organic matters [81], high concentrations of inorganic nutrients [82], and high EC [82,83]. However, they may be present in high levels in estuaries without showing any reduction in health [80]. *Nanivula perminuta*, another major species, exhibited different concentrations in ammonium and nitrate that showed a peak growth rate according to salt concentrations [84]. Therefore, to establish the biological impact assessment system using epilithic diatoms in water areas with severe fluctuation, such as estuaries, it is necessary to adjust the index values according to the nutritional status of the estuary instead of using freshwater indices as they are. Additional studies are also required to further understand the role of epilithic diatom communities.

## 5. Conclusions

Between 2009 and 2016 during May, we assessed the feasibility of applying diatom indices previously studied to assess the biological integrity of estuaries, while also predicting the importance of environmental factors and species appearance of epilithic diatoms in the southern part of the Korean Peninsula.

1. In total, 564 taxa of diatoms were found and the dominant species were identified as *Nitzschia inconspicua* and *N. perminuta*.

2. According to the species appearance and their abundances of diatoms, estuaries in the Korean Peninsula were geographically categorized into four groups. G1 showed high water temperature and DO levels, while nutrient levels were significantly low. G3 and G4 showed significantly high turbidity and nutrient levels.

3. A random forest model indicated that the major factors predicting diatom appearance in estuary are electric conductivity, salinity, turbidity, and total nitrogen.

4. The biological integrity of the estuary of Korean peninsula using “stream diatom indices” is very low through the sampling sites; however, a de novo diatom index should be developed to assess different or specialized ecosystem of estuary.

## Figures and Tables

**Figure 1 ijerph-16-02681-f001:**
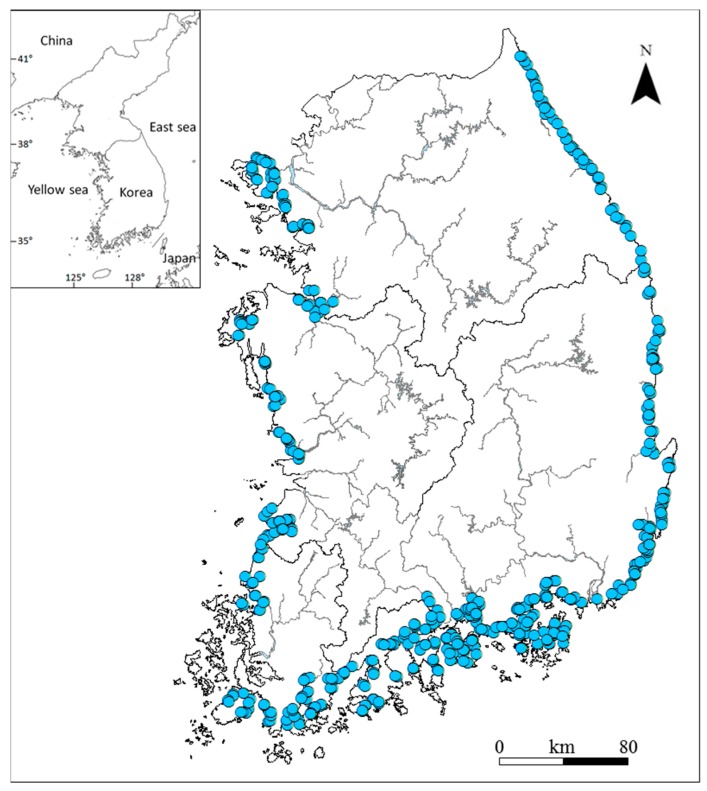
A map showing the 512 sampling stations for the water and diatom sampling in Korean estuaries between 2009 and 2016.

**Figure 2 ijerph-16-02681-f002:**
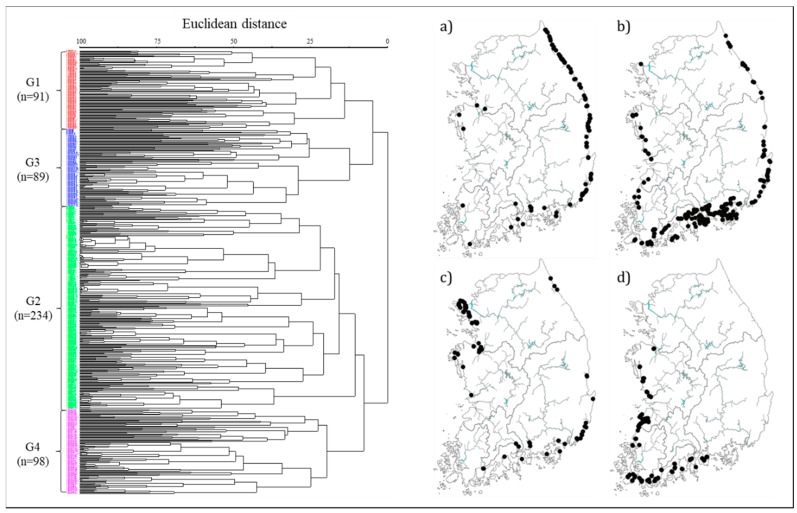
(**Left**) Dendrogram illustrating the sites affinity by cluster analysis with diatom abundance and appearance. Each color in figures presents four groups: G1 (red), G2 (green), G3 (blue), and G4 (pink). (**Right**) Geographical distribution patterns of the sampling site in four clusters: (**a**) G1; (**b**) G2; (**c**) G3; and (**d**) G4.

**Figure 3 ijerph-16-02681-f003:**
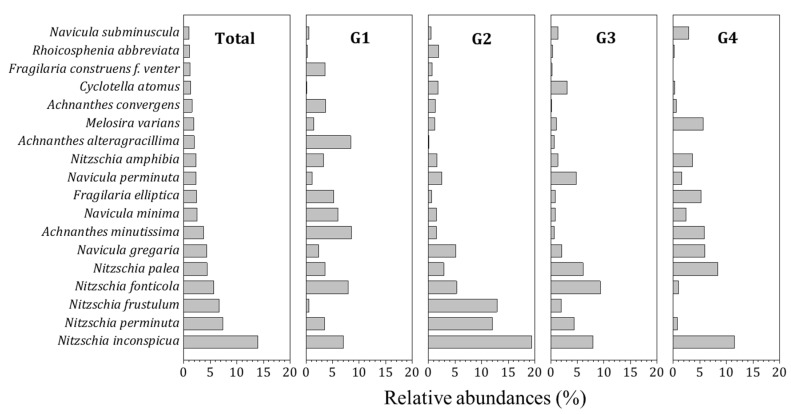
Relative abundances (%) of major epilithic diatom species observed in four groups in Korean estuaries between 2009 and 2016. The dominant species comprise those over the 0.5% of the total abundance in each group.

**Figure 4 ijerph-16-02681-f004:**
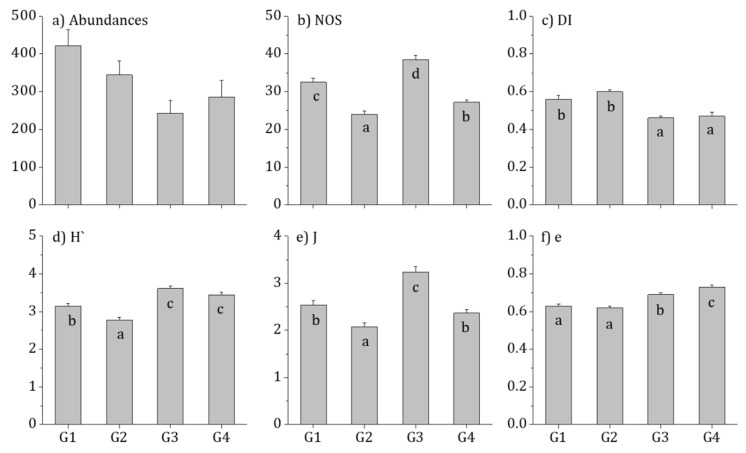
Biological factors in four groups of Korean estuaries between 2009 and 2016. Small letter (a–d) indicate Tukey’s post hoc test with the Bonferroni test: (**a**) abundances (1000 cells/cm^2^); (**b**) NOS, number of species; (**c**) DI, species dominant index; (**d**) H’, Shannon–Weaver’s diversity index; (**e**) J, richness; and (**f**) e, evenness.

**Figure 5 ijerph-16-02681-f005:**
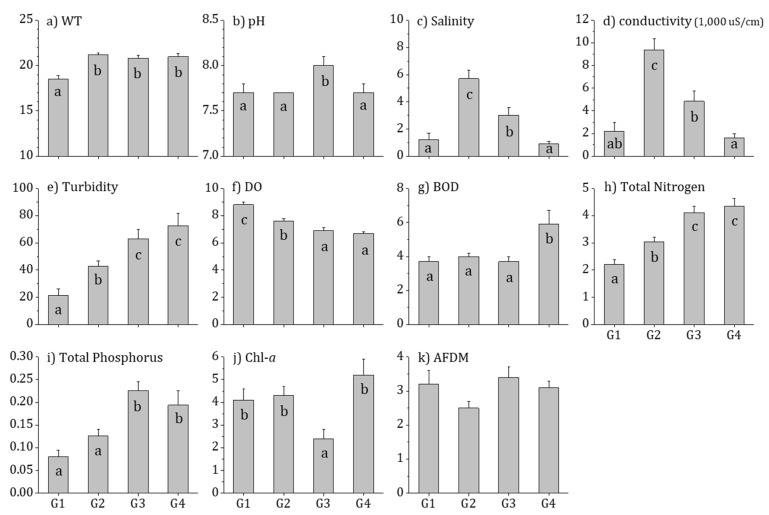
Environmental variables in four groups of Korean estuaries between 2009 and 2016. Small letters (a–c) indicate Tukey’s post hoc test with the Bonferroni test: (**a**) WT, Water temperature (°C); (**b**) pH; (**c**) Salinity (ppt); (**d**) conductivity, electric conductivity (1000 μS/cm); (**e**) Turbidity; (**f**) DO, Dissolve oxygen (mg/L); (**g**) BOD, Biochemical oxygen demand (mg/L); (**h**) Total Nitrogen (mg/L); (**i**) Total Phosphorus (mg/L); (**j**) Chl-*a*, Chlorophyll-*a* (mg/cm^2^); and (**k**) AFDM, Ash-free dry-matter (mg/cm^2^).

**Figure 6 ijerph-16-02681-f006:**
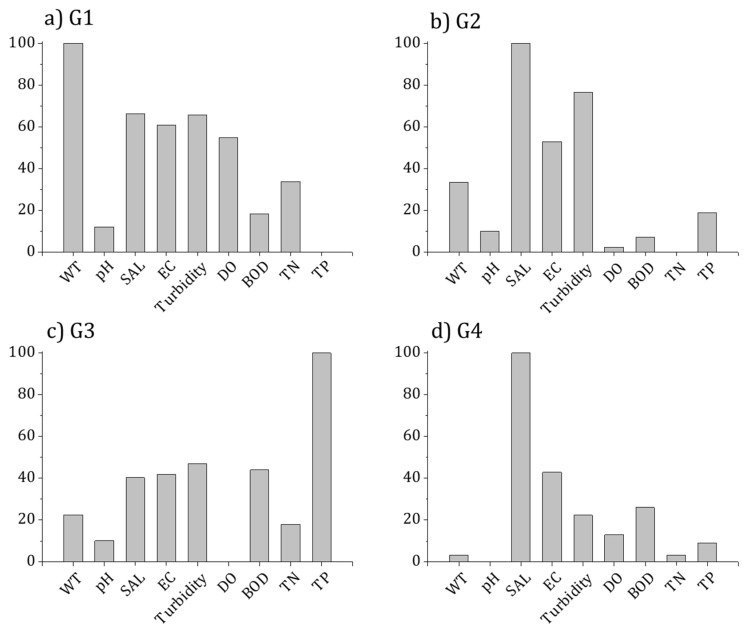
Relative importance (%) of predictable variables using a random forest model in four diatom communities defined by cluster analysis based on diatom abundance in Korean estuaries between 2009 and 2016: (**a**) G1; (**b**) G2; (**c**) G3; and (**d**) G4. WT, water temperature; SAL, salinity; EC, electric conductivity; DO, dissolved oxygen; BOD, biochemical oxygen demand; TN, total nitrogen; TP, total phosphorus.

**Figure 7 ijerph-16-02681-f007:**
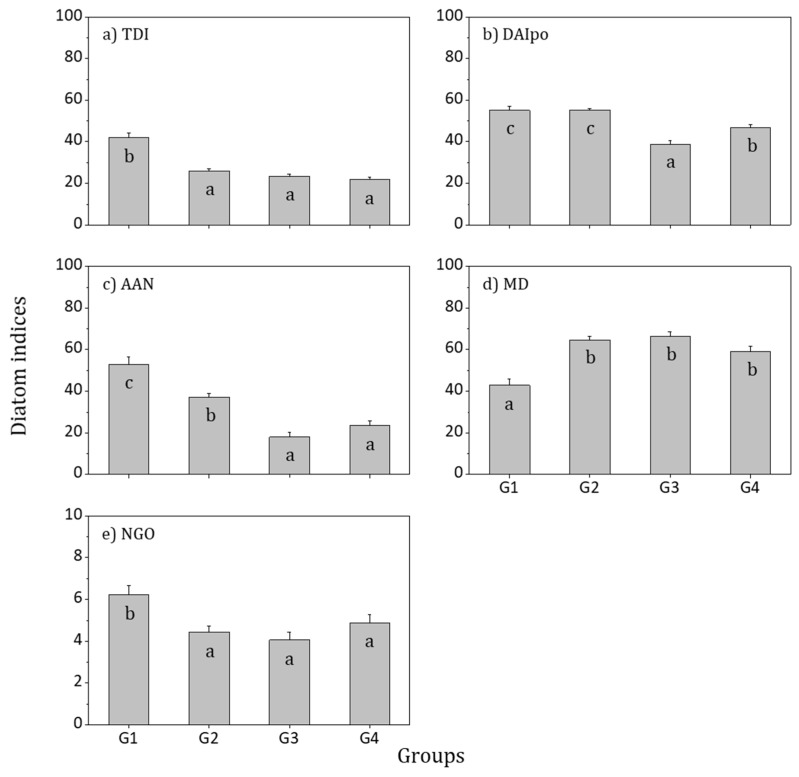
Relative diatom indices and biological water healthy of each epilithic diatom community in the Korean estuaries between 2009 and 2016. Small letters (a–c) indicate Tukey’s post hoc test with the Bonferroni test: (**a**) TDI, Trophic Diatom Index [51]; (**b**) DAIpo, Diatom Assemblage Index of Pollution [21]; (**c**) ANN (%), (*Achnanthes*/(*Achnanthes* + *Navicula*)) × 100 [52]; (**d**) MD (%), (motile diatom/total diatom abundance) × 100 [53]; and (**e**) NGO (%), (Number of *Gomphonema* species/Number of total diatom species) × 100 [52].

**Table 1 ijerph-16-02681-t001:** Collection sites of water and epilithic diatoms in the downstream or estuary of Korean peninsula between 2009 and 2016.

Locations	Open Stream *	Closed Stream **	Total
East sea	124	11	135
South sea	149	81	230
West sea	23	124	147

* Open streams have no dams to harvest the flowing water or to keep out the sea. ** Closed streams have one or several dams to harvest the flowing water or to keep out the sea.

**Table 2 ijerph-16-02681-t002:** Dominant species and number of species in Korean peninsula estuaries between 2009 and 2016.

Locations	Dominant Species (%)	Subdominant Species (%)	No. Species
East sea	*Nitzschia inconspicua* (8.8)	*Nitzschia fonticola* (8.3)	354
South sea	*Nitzschia inconspicua* (18.9)	*Nitzschia perminuta* (11.2)	457
West sea	*Nitzschia inconspicua* (10.2)	*Nitzschia palea* (6.8)	346
Total	*Nitzschia inconspicua* (13.9)	*Nitzschia perminuta* (7.4)	566

**Table 3 ijerph-16-02681-t003:** Good indicator species and the values (%) of the epilithic diatom communities in Korean peninsula estuaries between 2009 and 2016. Groups were divided by cluster analysis with diatom abundance and appearance.

Indicator Species	CODE	G1	G2	G3	G4	*p*
*Cymbella silesiaca*	CYSI	39	5	5	2	<0.001
*Fragilaria rumpens* var. *fragilarioides*	FRRV	30	0	5	0	<0.001
*Fragilaria capucina* var. *gracilis*	FRVG	28	0	5	0	<0.001
*Fragilaria construens* f. *venter*	FRCO	27	4	0	0	<0.001
*Reimeria sinuata*	RESI	27	1	2	0	<0.001
*Stephanodiscus invisitatus*	STIN	0	0	46	0	<0.001
*Cyclotella atomus*	CYAT	0	1	39	6	<0.001
*Stephanodiscus hantzschii*	STHA	0	0	38	3	<0.001
*Nitzschia constricta*	NICN	0	1	35	0	<0.001
*Navicula atomus*	NAAT	1	0	34	0	<0.001
*Navicula veneta*	NAVE	5	0	33	2	<0.001
*Navicula halophila*	NAHA	1	0	32	0	<0.001
*Navicula accomoda*	NAAC	1	1	25	1	<0.001
*Bacillaria paradoxa*	BAPA	1	2	6	37	<0.001
*Navicula capitata*	NACA	0	0	3	30	<0.001
*Nitzschia calida*	NICA	0	1	1	25	<0.001

**Table 4 ijerph-16-02681-t004:** Relationship between indicator species and environmental variables in each diatom group in Korean estuaries between 2009 and 2016. Groups were divided by cluster analysis with diatom abundance and appearance. CODE (name of diatom) cited from Table 3.

CODE	WT	pH	SAL	EC	TURB	DO	BOD	TN	TP	CHL	AFDM	G
CYSI	−0.306 **	−0.130 **	−0.212 **	−0.270 **	−0.319 **	0.274 **	−0.137 **	−0.225 **	−0.194 **	0.064	−0.160 **	1
FRRV	−0.225 **	−0.021	−0.175 **	−0.185 **	−0.071	0.162 **	−0.039	−0.018	−0.035	−0.061	−0.054	1
FRVG	−0.217 **	−0.013	−0.196 **	−0.235 **	−0.097 *	0.159 **	−0.016	−0.022	−0.084	−0.088 *	−0.105 *	1
FRCO	−0.149 **	−0.074	−0.106 *	−0.056	−0.164 **	0.199 **	−0.031	−0.152 **	−0.078	0.074	0.127 **	1
RESI	−0.106 *	−0.074	−0.143 **	−0.210 **	−0.145 **	0.183 **	−0.130 **	−0.114 **	−0.124 **	−0.007	−0.094 *	1
STIN	0.018	0.160 **	−0.005	0.053	0.196 **	−0.062	−0.018	0.170 **	0.183 **	−0.203 **	0.141 **	3
CYAT	0.130 **	0.211 **	0.019	0.059	0.319 **	−0.226 **	−0.068	0.188 **	0.162 **	−0.275 **	0.230 **	3
STHA	0.009	0.174 **	−0.033	0.025	0.261 **	−0.123 **	0.073	0.208 **	0.198 **	−0.214 **	0.164 **	3
NICN	0.039	0.055	0.123 **	0.147 **	0.082	−0.005	−0.014	−0.032	0.039	−0.086	0.053	3
NAAT	−0.018	0.158 **	−0.095 *	−0.049	0.189 **	−0.179 **	−0.075	0.187 **	0.217 **	−0.202 **	0.157 **	3
NAVE	−0.023	0.089 *	−0.157 **	−0.082	0.151 **	−0.016	0.042	0.172 **	0.109 *	−0.185 **	0.075	3
NAHA	0.002	0.088 *	0.007	0.068	0.147 **	−0.091 *	−0.016	0.126 **	0.158 **	−0.187 **	0.147 **	3
NAAC	−0.042	0.100 *	0.025	−0.015	0.154 **	−0.021	−0.022	−0.018	0.112 *	−0.011	0.066	3
BAPA	0.087	0.042	−0.024	0.060	0.137 **	−0.042	0.144 **	0.024	0.005	0.061	0.093 *	4
NACA	0.065	0.062	−0.102 *	−0.074	0.196 **	−0.072	0.135 **	0.107 *	0.079	0.055	0.044	4
NICA	0.090 *	0.006	0.058	0.073	0.106 *	−0.160 **	0.101 *	0.084	0.074	0.077	0.070	4

WT, water temperature; EC, electric conductivity; TURB, turbidity; DO, dissolved oxygen; BOD, biochemical oxygen demand; TN, total nitrogen; TP, total phosphorus; CHL, chlorophyll-a; AFDM, ash-free-dry matter; G, group. * *p* < 0.05, ** *p* < 0.01.

**Table 5 ijerph-16-02681-t005:** List of epilithic diatoms with the first and second most important variables predicting species appearance by a random forest model in Korean estuaries between 2009 and 2016. Ar, accuracy rate; AUC, area under the curve; GI, group indicator.

Species	Ar	AUC	Important Variables		GI
			1st	2nd	
*Achnanthes alteragracillima* Lange-Bertalot	0.88	0.99	TN (100)	TURB (94)	
*Achnanthes brevipes* Agardh	0.88	0.98	SAL (100)	EC (75)	
*Achnanthes brevipes* var. *intermedia* (Kützing) Cleve	0.93	0.98	EC (100)	TURB (61)	
*Achnanthes clevei* Grunow	0.94	0.99	TN (100)	EC (90)	
*Achnanthes conspicua* A. Mayer	0.95	0.99	TP (100)	TN (93)	
*Achnanthes convergens* Kobayasi, Nagumo & Mayama	0.85	0.94	TN (100)	SAL (97)	
*Achnanthes delicatula* (Kützing) Grunow	0.83	0.99	WT (100)	TP (96)	
*Achnanthes exigua* Grunow	0.84	0.96	DO (100)	TURB (99)	
*Achnanthes hungarica* (Grunow) Grunow	0.93	0.99	SAL (100)	EC (88)	
*Achnanthes inflata* (Kützing) Grunow	0.94	1.00	SAL (100)	WT (84)	
*Achnanthes lanceolata* (Brébisson) Grunow	0.90	0.96	EC (100)	TURB (97)	
*Achnanthes laterostrata* Hustedt	0.93	0.99	pH (100)	TN (92)	
*Achnanthes minutissima* Kützing	0.86	0.95	EC (100)	TP (51)	
*Achnanthes subhudsonis* Hustedt	0.91	0.96	TURB (100)	EC (95)	
*Amphora* sp.	0.93	0.99	TN (100)	EC (97)	
*Amphora coffeaeformis* (Agardh) Kützing	0.90	0.97	SAL (100)	EC (54)	
*Amphora copulate* (Kützing) Schoeman & Archibald	0.90	0.98	EC (100)	TP (65)	
*Amphora montana* Krasske	0.93	0.99	BOD (100)	TURB (78)	
*Amphora pediculus* (Kützing) Grunow	0.90	0.96	BOD (100)	TP (48)	
*Amphora veneta* Kützing	0.92	0.99	EC (100)	SAL (71)	
*Aulacoseira alpigena* (Grunow) Krammer	0.93	0.99	EC (100)	SAL (65)	
*Aulacoseira ambigua* (Grunow) Simonsen	0.90	0.97	TP (100)	pH (42)	
*Aulacoseira granulata* (Ehrenberg) Simonsen	0.92	0.98	EC (100)	pH (85)	
*Bacillaria paradoxa* Gmelin	0.85	0.99	BOD (100)	EC (92)	4
*Cocconeis placentula* Ehrenberg	0.86	0.99	SAL (100)	EC (76)	
*Cocconeis placentula* var. *euglypta* (Ehrenberg) Grunow	0.91	0.97	TN (100)	SAL (99)	
*Cocconeis placentula* var. *lineata* (Ehrenberg) Van Heurck	0.88	0.97	EC (100)	TURB (77)	
*Cyclostephanos dubius* (Hustedt) Round	0.95	0.99	pH (100)	EC (52)	
*Cyclotella atomus* Hustedt	0.92	0.98	TURB (100)	WT (40)	3
*Cyclotella meneghiniana* Kützing	0.83	0.96	TURB (100)	TP (77)	
*Cyclotella pseudostelligera* Hustedt	0.84	0.98	TP (100)	WT (80)	
*Cyclotella stelligera* (Cleve & Grunow) Van Heurck	0.92	0.99	TURB (100)	TP (58)	
*Cymbella affinis* Kützing	0.88	0.97	EC (100)	SAL (74)	
*Cymbella minuta* Hilse	0.86	0.97	EC (100)	SAL (80)	
*Cymbella silesiaca* Bleisch	0.87	0.96	EC (100)	TURB (97)	1
*Cymbella tumida* (Brébisson) Van Heurck	0.91	0.96	TURB (100)	TP (59)	
*Diatoma vulgaris* Bory	0.95	0.98	EC (100)	SAL (87)	
*Diploneis oblongella* (Nägeli ex Kützing) Cleve-Euler	0.95	0.99	SAL (100)	BOD (88)	
*Diploneis subovalis* Cleve	0.87	0.98	SAL (100)	EC (75)	
*Entomoneis alata* (Ehrenberg) Ehrenberg	0.94	0.98	EC (100)	SAL (99)	
*Eunotia minor* (Kützing) Grunow	0.95	1.00	EC (100)	TN (94)	
*Fragilaria capitellata* (Grunow) J.B. Petersen	0.90	0.96	WT (100)	TURB (92)	
*Fragilaria capucina* Desmazières	0.83	0.95	EC (100)	WT (59)	
*Fragilaria capucina* var. *gracilis* (Østrup) Hustedt	0.90	0.96	SAL (100)	TN (48)	1
*Fragilaria capucina* var. *vaucheriae* (Kützing) Lange-Bertalot	0.98	0.99	TURB (100)	TP (40)	
*Fragilaria construens* f. *venter* (Ehrenberg) Hustedt	0.89	0.98	SAL (100)	TURB (74)	1
*Fragilaria elliptica* Schumann	0.90	0.97	TURB (100)	WT (76)	
*Fragilaria fasciculata* (Agardh) Lange-Bertalot	0.93	0.99	SAL (100)	EC (83)	
*Fragilaria parva* (Grunow) A. Tuji & D.M. Williams	0.93	0.99	SAL (100)	DO (95)	
*Fragilaria pinnata* Ehrenberg	0.86	0.98	WT (100)	SAL (85)	
*Fragilaria rumpens* (Kützing) G.W.F. Carlson	0.88	0.97	SAL (100)	WT (68)	
*Fragilaria rumpens* var. *familiaris* (Kützing) Grunow	0.87	0.97	SAL (100)	EC (63)	
*Fragilaria rumpens* var. *fragilarioides* (Grunow) Cleve	0.91	0.97	WT (100)	SAL (51)	1
*Frustulia vulgaris* (Thwaites) De Toni	0.94	0.99	EC (100)	BOD (73)	
*Gomphonema angustum* Agardh	0.94	0.98	WT (100)	BOD (91)	
*Gomphonema clevei* Fricke	0.87	0.96	EC (100)	DO (25)	
*Gomphonema lagenula* Kützing	0.82	0.98	EC (100)	SAL (72)	
*Gomphonema minutum* (Agardh) Agardh	0.94	0.99	WT (100)	TP (78)	
*Gomphonema parvulum* (Kützing) Kützing	0.86	0.98	TN (100)	SAL (66)	
*Gomphonema pseudoaugur* Lange-Bertalot	0.88	0.97	TP (100)	SAL (99)	
*Gomphonema quadripunctatum* (Østrup) Wislouch	0.92	0.96	pH (100)	BOD (81)	
*Gomphonema truncatum* Ehrenberg	0.94	0.99	SAL (100)	EC (86)	
*Gyrosigma acuminatum* (Kützing) Rabenhorst	0.91	0.99	EC (100)	DO (84)	
*Hantzschia amphioxys* (Ehrenberg) Grunow	0.91	0.98	SAL (100)	EC (68)	
*Melosira nummuloides* Agardh	0.94	0.99	TURB (100)	TP (69)	
*Melosira varians* Agardh	0.91	0.98	EC (100)	TP (92)	
*Meridion circulare* var. *constrictum* (Ralfs) Van Heurck	0.92	0.98	EC (100)	BOD (81)	
*Navicula accomoda* Hustedt	0.88	0.98	TURB (100)	WT (71)	3
*Navicula atomus* (Kützing) Grunow	0.96	0.99	SAL (100)	BOD (71)	3
*Navicula atomus* var. *permitis* (Hustedt) Lange-Bertalot	0.98	1.00	TURB (100)	SAL (99)	
*Navicula bacillum* Ehrenberg	0.95	0.99	pH (100)	SAL (67)	
*Navicula capitate* (Ehrenberg) R. Ross	0.87	0.97	SAL (100)	DO (97)	4
*Navicula capitatoradiata* H. Germain	0.92	0.97	SAL (100)	EC (84)	
*Navicula cincta* (Ehrenberg) Ralfs	0.91	1.00	SAL (100)	EC (93)	
*Navicula clementis* Grunow	0.93	0.98	TN (100)	TP (58)	
*Navicula contenta* Grunow	0.90	0.98	SAL (100)	EC (74)	
*Navicula cryptocephala* Kützing	0.87	0.96	EC (100)	SAL (61)	
*Navicula cryptotenella* Lange-Bertalot	0.83	0.97	EC (100)	WT (70)	
*Navicula decussis* Østrup	0.91	0.97	EC (100)	SAL (48)	
*Navicula goeppertiana* (Bleisch) H.L. Smith	0.88	0.98	EC (100)	TURB (74)	
*Navicula gregaria* Donkin	0.82	0.97	SAL (100)	EC (77)	
*Navicula halophile* (Grunow) Cleve	0.95	0.98	EC (100)	SAL (91)	3
*Navicula menisculus* Schumann	0.93	1.00	EC (100)	SAL (86)	
*Navicula minima* Grunow	0.88	0.96	EC (100)	SAL (71)	
*Navicula minuscula* Grunow	0.88	0.98	TP (100)	SAL (98)	
*Navicula mutica* (Kützing) Frenguelli	0.92	0.99	EC (100)	SAL (84)	
*Navicula mutica* var. *ventricosa* (Kützing) Cleve & Grunow	0.92	0.99	SAL (100)	EC (54)	
*Navicula peregrine* (Ehrenberg) Kützing	0.95	1.00	EC (100)	SAL (87)	
*Navicula perminuta* Grunow	0.92	0.98	WT (100)	pH (40)	
*Navicula phyllepta* Kützing	0.94	0.99	WT (100)	BOD (87)	
*Navicula pupula* Kützing	0.83	0.97	TN (100)	SAL (84)	
*Navicula radiosa* Kützing	0.89	0.98	TN (100)	pH (54)	
*Navicula recens* (Lange-Bertalot) Lange-Bertalot	0.84	0.98	EC (100)	SAL (87)	
*Navicula rhynchocephala* Kützing	0.91	1.00	BOD (100)	EC (66)	
*Navicula salinarum* Grunow	0.86	0.97	TURB (100)	TP (73)	
*Navicula saprophila* Lange-Bertalot & Bonik	0.90	0.98	TP (100)	WT (88)	
*Navicula schroeteri* F. Meister	0.89	0.98	TN (100)	SAL (99)	
*Navicula seminuloides* Hustedt	0.95	0.99	TP (100)	pH (60)	
*Navicula seminulum* Grunow	0.93	0.97	EC (100)	SAL (83)	
*Navicula subatomoides* Hustedt	0.92	0.98	TN (100)	TP (86)	
*Navicula subminuscula* Manguin	0.85	0.95	SAL (100)	TP (68)	
*Navicula tenera* Hustedt	0.93	1.00	SAL (100)	EC (89)	
*Navicula tripunctata* (O.F. Müller) Bory	0.91	0.97	SAL (100)	EC (58)	
*Navicula trivialis* Lange-Bertalot	0.87	0.98	SAL (100)	EC (49)	
*Navicula veneta* Kützing	0.88	0.97	TP (100)	EC (56)	3
*Navicula viridula* (Kützing) Ehrenberg	0.95	0.98	EC (100)	SAL (73)	
*Navicula viridula* var. *rostellata* (Kützing) Cleve	0.86	0.98	SAL (100)	EC (56)	
*Nitzschia acicularis* (Kützing) W. Smith	0.94	0.98	DO (100)	WT (94)	
*Nitzschia amphibia* Grunow	0.90	0.97	WT (100)	SAL (96)	
*Nitzschia calida* Grunow	0.89	0.99	DO (100)	BOD (85)	4
*Nitzschia capitellata* Hustedt	0.87	0.99	BOD (100)	TP (75)	
*Nitzschia communis* Rabenhorst	0.91	0.99	EC (100)	SAL (72)	
*Nitzschia constricta* (Kützing) Ralfs	0.88	0.97	EC (100)	SAL (54)	3
*Nitzschia dissipata* (Kützing) Rabenhorst	0.82	0.99	SAL (100)	TP (58)	
*Nitzschia filiformis* (W. Smith) Van Heurck	0.86	0.98	SAL (100)	pH (92)	
*Nitzschia fonticola* (Grunow) Grunow	0.88	0.97	SAL (100)	DO (54)	
*Nitzschia frustulum* (Kützing) Grunow	0.89	0.99	SAL (100)	EC (91)	
*Nitzschia gracilis* Hantzsch	0.87	0.96	TN (100)	BOD (75)	
*Nitzschia inconspicua* Grunow	0.83	0.97	EC (100)	TN (91)	
*Nitzschia linearis* W. Smith	0.89	0.98	WT (100)	TP (68)	
*Nitzschia littoralis* Grunow	0.91	0.99	WT (100)	EC (77)	
*Nitzschia nana* Grunow	0.94	0.99	EC (100)	BOD (90)	
*Nitzschia palea* (Kützing) W. Smith	0.86	0.97	SAL (100)	EC (95)	
*Nitzschia paleacea* (Grunow) Grunow	0.84	0.98	BOD (100)	SAL (44)	
*Nitzschia pellucida* Grunow	0.95	0.99	EC (100)	SAL (89)	
*Nitzschia perminuta* (Grunow) M. Peragallo	0.92	0.98	TURB (100)	pH (98)	
*Nitzschia tryblionella* Hantzsch	0.95	0.99	TN (100)	DO (92)	
*Reimeria sinuata* (W. Gregory) Kociolek & Stoermer	0.92	0.97	EC (100)	TURB (45)	1
*Rhoicosphenia abbreviate* (Agardh) Lange-Bertalot	0.92	0.97	TURB (100)	TP (21)	
*Stephanodiscus hantzschii* Grunow	0.94	0.98	pH (100)	TURB (63)	3
*Stephanodiscus invisitatus* Hohn & Hellermann	0.94	0.98	pH (100)	TP (64)	3
*Surirella angusta* Kützing	0.90	0.97	EC (100)	SAL (78)	
*Surirella minuta* Brébisson ex Kützing	0.83	0.97	SAL (100)	EC (82)	
*Surirella ovalis* Brébisson	0.94	0.99	TP (100)	WT (65)	
*Surirella ovata* Kützing	0.96	0.99	TP (100)	TURB (90)	
*Synedra acus* Kützing	0.86	0.99	SAL (100)	BOD (79)	
*Synedra pulchella* (Ralfs ex Kützing) Kützing	0.90	0.99	SAL (100)	EC (95)	
*Synedra ulna* (Nitzsch) Ehrenberg	0.86	0.96	SAL (100)	EC (83)	
*Thalassiosira bramaputrae* (Ehrenberg) Håkansson & Locker	0.91	0.99	EC (100)	TURB (91)	

WT, water temperature; SAL, salinity; DO, dissolved oxygen; BOD, biochemical oxygen demand; TURB, turbidity; EC, electric conductivity; TN, total nitrogen; TP, total phosphorus.

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
