# Peer review of "Benthic Diatom Communities in Korean Estuaries: Species Appearances in Relation to Environmental Variables"

_ijerph, 2019, doi:10.3390/ijerph16152681_

Round 1

Reviewer 1 Report

In the chapter of the conclusion should explain how the investigated area is different and/or similar comparing with other similar water types. It is also important to emphasize what the new for the science in the manuscript.

Author Response

Thank you very much. 

We revised the conclusion as following sentences:

Between 2009 and 2016 during May, we assessed the feasibility of applying diatom indices previously studied to assess the biological integrity of estuaries, while also predicting the importance of environmental factors and species appearance of epilithic diatoms in the southern part of the Korean Peninsula.

1. A total of 564 taxa of diatoms were found and the dominant species were identified as Nitzschia inconspicua and N. perminuta.

2. According to the species appearance and their abundances of diatoms, estuaries in the Korean Peninsula were geographically categorized into four groups. G1 showed high water temperature and DO levels, while nutrient levels were significantly low. Meanwhile, G3 and G4 showed significantly high turbidity and nutrient levels.

3. A random forest model indicate that major factors predicting diatom appearance in estuary are electric conductivity, salinity, turbidity, and total nitrogen.

4. The biological integrity of the estuary of Korean peninsula using stream diatom indices’ is very low through the sampling sites, but a de novo diatom index should be developed to assess different or specific ecosystem of estuary.

Reviewer 2 Report

This is a very relevant paper regarding environmental issues in the Korean Peninsula. It is an application of basic statistical methodologies but, despite being simple, it is ideal for obtaining relevant environmental results, which are indeed interesting and may provide some guidance or inspirations for environmental management.

Relevant Points:

1)      The sampling design: How did the authors choose the 512 sampling stations? This choice should be further explained. I have doubts regarding randomness and representativeness of the sample.

2)      The paper involves a high dimension of variables (for instance, taxa of epilithic diatoms is 566). There should be a subsection in Section 2.3 Data Analysis to summarize the variables, some basic descriptive statistics of the variables, particularly the sample sizes (n=?). This would help to simplify the reading of the paper and to better understand the issue under study.

3)      Lines 78-79: “The surveys were conducted in May to avoid the influence of the heavy rainy season during summer.” In that month, how many surveys were conducted: a single time in that month and in each of 512 estuarine zones? Data periodicity and dimension are not well presented.

4)      Table 3: in P column, values “0.001”. Are all values the same? Shouldn’t it be instead: <0.001?

5)      Line 237: “5.7%º” should be “5.7%” (the whole text should be read to detect such minor spelling errors)

6)      Legends of Figures 4, 5, 7: The authors wrote “Small alphabets 225 (a, b, and c)”, but in the bar chart also appears letter “d” (see Figure 4, for instance). This should be confirmed.

Author Response

This is a very relevant paper regarding environmental issues in the Korean Peninsula. It is an application of basic statistical methodologies but, despite being simple, it is ideal for obtaining relevant environmental results, which are indeed interesting and may provide some guidance or inspirations for environmental management.

ð  Thank you very much for you kind comments.

Relevant Points:

The sampling design: How did the authors choose the 512 sampling stations? This choice should be further explained. I have doubts regarding randomness and representativeness of the sample.

ð  Thank you for your question. We inserted the following sentence; The selected sites are an actual total survey points designated by Ministry of Environment (MOE), Republic of Korea. Therefore, we selected all the river mouth (estuary) for the study.

The paper involves a high dimension of variables (for instance, taxa of epilithic diatoms is 566). There should be a subsection in Section 2.3 Data Analysis to summarize the variables, some basic descriptive statistics of the variables, particularly the sample sizes (n=?). This would help to simplify the reading of the paper and to better understand the issue under study.

ð  Thank you for your advices. We made a three subsection to better understand.

Lines 78-79: “The surveys were conducted in May to avoid the influence of the heavy rainy season during summer.” In that month, how many surveys were conducted: a single time in that month and in each of 512 estuarine zones? Data periodicity and dimension are not well presented.

ð  Thank you for your question. We inserted the following sentence; The 512 locations are not regular and unrepeated samples or points surveyed, just irregularly sampled in May during 2009-2016. In Korea, May has little or no impact of monsoon or rainfall against stream and estuary ecosystem.

Table 3: in P column, values “0.001”. Are all values the same? Shouldn’t it be instead: <0.001?

ð  Thank you very much. You are alright. We revised as <0.001 for all in Table.

Line 237: “5.7%º” should be “5.7%” (the whole text should be read to detect such minor spelling errors)

ð  Thank you very much. We exchanged with unit “ppt”, not %.

Legends of Figures 4, 5, 7: The authors wrote “Small alphabets 225 (a, b, and c)”, but in the bar chart also appears letter “d” (see Figure 4, for instance). This should be confirmed.

ð  Thank you very much. According to reviewer’s comment, we inserted the unit “d”, then a, b, c and d.